# UV Fingerprinting Approaches for Quality Control Analyses of Food and Functional Food Coupled to Chemometrics: A Comprehensive Analysis of Novel Trends and Applications

**DOI:** 10.3390/foods11182867

**Published:** 2022-09-16

**Authors:** Mohamed A. Farag, Mohamed Sheashea, Chao Zhao, Amal A. Maamoun

**Affiliations:** 1Pharmacognosy Department, College of Pharmacy, Cairo University, Kasr el Aini St., Cairo P.B. 11562, Egypt; 2Aromatic and Medicinal Plants Department, Desert Research Center, Cairo P.B. 11435, Egypt; 3College of Marine Sciences, Fujian Agriculture and Forestry University, Fuzhou 350002, China; 4Engineering Research Centre of Fujian-Taiwan Special Marine Food Processing and Nutrition, Ministry of Education, Fuzhou 350002, China; 5Pharmacognosy Department, National Research Centre, 33 Elbohouth St., Dokki, Giza 12622, Egypt

**Keywords:** UV spectroscopy, chemometrics, food analysis, nutraceuticals, quality control

## Abstract

(1) Background: Ultraviolet-visible (UV-Vis) spectroscopy is a common analytical tool to detect chromophore in compounds by monitoring absorbance spectral wavelengths. Further, it could provide spectral information about complex conjugated systems in mixtures aided by chemometric tools to visualize large UV-Vis datasets as typical in food samples. This review provides novel insight on UV-Vis applications in the last 20 years, as an advanced analytical tool in the quality control of food and dietary supplements, as well as several other applications, including chemotaxonomy, authentication, fingerprinting, and stability studies. (2) Conclusions: A critical assessment of the value of UV application and its novel trends in the quality control (QC) of nutraceuticals reveals the advantages and limitations, focusing on areas where future advancements are in need. Although simple, UV and its novel trends present potential analytical tools with an acceptable error for QC applications from a non-targeted perspective compared to other expensive spectral tools.

## 1. Introduction

The ultraviolet (UV) spectrometer is well-recognized in chemical assays as one of the most common analytical devices to provide accuracy, rapidity, simplicity, and low cost compared to other analytical platforms. UV-Vis spectroscopy is an analytical technique used to measure the number of discrete wavelengths of UV “expressed in nanometers” absorbed/transmitted by samples affected by their composition and concentration. The maxima (λ_max_) and intensities of wavelengths’ absorption differ relative to the molecular structure of the compound vs. chromophores, and auxochromes, in addition to changes in sample level, with the lesser effect of other impurities present [1].

Natural products and their derivatives are well-recognized as potential sources for biologically active substances. The existence of natural products in nature in such complex biodiversity warrants advanced analytical tools and resources for its bioassay. UV spectroscopy is one of the most common spectroscopic tools commonly used for identification and quantification purposes [2]. Application of UV-Vis in the analysis of nutraceuticals or herbal drugs has recently emerged, including quality control determination, interspecies curvature, certification, and adulteration detection based primarily on UV-Vis fingerprinting [3].

Considering the complexity of more dense spectral datasets typically originating from several samples, the coupling of UV-Vis spectra with chemometric tools appears to be powerful, revealing data classification among specimens, and in some cases aiding in the identification of markers in a non-targeted manner [3]. The outcomes of the combination of UV-Vis spectra and chemometrics were revealed in many plant studies that assessed the effect of different cultivars or geographical origins on the composition of metabolites [4].

Application of UV-Vis has also encompassed food analysis, exemplified in the determination of caffeine in beverages [5] and anthocyanin levels in wine by monitoring caffeine and anthocyanins‘ distinct UV absorption peaks maxima, respectively [6]. In industry, UV applications extend to detect the presence or removal of dyes by comparing post-treatment UV absorption peaks [7]. In addition to monitoring changes in UV absorption wavelengths to detect changes in protein structures [8], advanced applications characterize tiny nanoparticles depending on shifts in peak absorbance λ_max_ [9].

Using UV-Vis analysis of metabolites as a tool to answer these questions in foods, functional foods QC typically leads to relatively complex datasets, and in addition to monitoring wavelengths, many sample observations are used as bioinformatics tools to help them dissipate variables. Chemometric tools used in quality control (QC) handling large datasets of UV spectrum are attributed to multivariate regression methods, multivariate decomposition methods, hierarchical cluster analysis, and pattern recognition methods. The relationship between samples and variables in a given UV dataset is revealed by using suitable tools. Multivariate regression methods for calibration and prediction are selected based on the complexity of the datasets, with the advantageous application of multivariate calibration over regression is processing multiple variables’ interaction of analytes with each other, especially in the QC of herbal products [10]. Further, multivariate decomposition methods used to resolve multivariate datasets by minimizing their dimensionality into some correlated variables are classified into principal component analysis (PCA) and parallel factor analysis (PARAFAC). Among multivariate-based modeling, PCA is considered the most widespread tool applicable in herbal and biological mixtures requiring correlated variables for significant results [11]. Moving to hierarchical cluster analysis (HCA), this type of multivariate analysis classifies components based on their similarity, exhibiting homogeneity within intergroup and heterogeneity among different groups suitable for chemotaxonomy and fingerprinting applications [12]. Other tools for QC, linear discrimination analysis (LDA) and soft independent modeling of class analogy (SIMCA), are involved in the search for a linear function between the variables of datasets grouped as pattern recognition methods [13]. The most common chemometric tools employed for QC analysis and discussed in this review include PCA, PLS, HCA, and LDA [14]. 

Such an approach combining UV-Vis spectra and chemometric tools has been increasingly used to resolve metabolome and allow for the detection of adulterants, assess for variability in botanicals [15], and to present the main part of this study. The next subsections will explain in detail different UV fingerprint-based applications and their advantages and/or limitations towards the analysis of foods and nutraceuticals, with the potential future uses and developments required highlighted. 

The principal domains of applications, the number of publications for each domain, and the corresponding percentage from the total number of publications in the UV-Vis fingerprinting area are presented in Figure 1. 

## 2. UV Applications in Food Quality Control, Authentication, Fingerprinting, and Assays

### 2.1. Fruits, Vegetables, and Spices Analyses

The development of simple, accurate, and rapid analytical methods for food authentication is warranted to ensure food quality and safety, especially if the food industry is used for quick food examination to ensure the best quality measures. In that context, an application of UV in food authentication includes the work of Fernández-León et al. (2010), which aimed to quantify chlorophyll A and B in broccoli and cabbage as markers of the photosynthetic membrane for analysis using UV spectra coupling (PLS). The limit of detection was at 0.174 and 0.304 μg mL^−1^; selectivity at 0.946 and 0.942; and sensitivity at 0.0324 and 0.0183 μg mL^−1^ for chlorophyll A and B, respectively. The validated PLS model was established for chlorophyll A and B levels in broccoli and cabbage cultivars and was further compared to the HPLC chromatographic technique and photometry based on two different wavelength measurements at 662 and 645 nm, respectively, to prove its significant improvement in both accuracy and precision [16]. 

Total carotenoids’ UV-Vis-based estimations (λ_max_ 450 nm) in different dietary sources, including pumpkin pulp, pumpkin juice, sea buckthorn fruits, sea buckthorn juice, and two corn hybrids, using two other extraction methods, were examined to determine carotenoids’ maximum extractability. The results of optimum extraction conditions of carotenoids from various dietary sources revealed that extraction using methanol/ethyl acetate/petroleum ether was more adequate for recovery from pumpkin pulp and juice. At the same time, for corn samples, the cold acetone method appeared more efficient. Furthermore, in the case of sea buckthorn samples, both extraction methods showed comparable results [17].

One application of UV-B (260–320 nm) irradiation assessed the impact of mutation “according to the method reported by [18]” in two genotypes of tomatoes named “Money maker and high pigment-1” at two different maturity stages, “mature green and turning”. Major metabolites “ascorbic acid and carotenoids” were evaluated after mutation to account for tomato nutritive value, which revealed a major increment in metabolites’ levels in Money maker flesh and peels versus a minor change in levels detected in high pigment-1. Although both genotype stages have a negative effect on fruit firmness after radiation, an increase in softness was detected [19].

UV-Vis coupled to chemometrics was employed to classify ten cassava genotypes (*Manihot esculenta*), where *β*-carotene was identified as a distinguishable biomarker at a maximum of 450 nm UV [20]. *β*-carotene is close to the visible region, away from the typical solvent wavelength cutoffs, as a potential marker for the discrimination of cassava specimens, and has yet to be used for its QC scheme. 

Qualitative determination of capsaicin in habanero peppers (Chinese capsicum) was reported using UV spectroscopy with PLS multivariate regression modeling conjugation at 284 nm. PLS was built by correlating the known total levels of the two major capsaicin (capsaicin and dihydrocapsaicin) in the associated extract, determined by UV spectral datasets without prior separation [21]. 

The metabolite profiling approach attempts to classify and identify various organisms, mainly plants, based upon their chemical similarities or differences. In this regard, chemical variability serves as a biomarker for chemotaxonomy relatedness of genotypes, where different genotypes could be discriminated according to their UV fingerprint. Genetic variability did not distinguish between cultivars or varieties warrant, exploiting metabolite fingerprints for their classification and also helping to detect other substitutes for official drugs [22]. 

For example, different *Curcuma* species from Indonesia, i.e., *C. longa, C. xanthorrhiza, C. aeruginosa*, and *C. mangga* were discriminated based on their UV-Vis spectra at 210–500 nm, analyzed using PCA and DA analysis. PCA analysis showed segregation of *C. longa* from other species due to its enrichment in curcuminoids, as revealed from the loading plot, whereas *C. aeruginosa* and *C. mangga* overlapped due to their related chemical profile. To overcome overlapping, discriminant analysis (DA) was used to discriminate between samples where *C. aeruginosa* and *C. mangga* were determined in the positive value of second discriminant function (DF2). The developed DA model allowed the classification of species from different samples with a 95.5% discrimination value [4]. 

Regarding adulterants’ detection in chili spice, UV-Vis spectroscopy and PCA modeling aided in the detection of adulterants such as rhodamine B and commercial red textile dye in chili commercial powder at 600 nm [23]. Likewise, UV-Vis coupled to PLS-DA was capable of detecting synthetic dye adulterants, i.e., Sudan I or blends of Sudan I + IV dyes in culinary spices [24]. 

Another application of spices’ authentication using UV-Vis fingerprint by Hegazi et al. (2022) discriminated metabolites of genuine saffron from its common adulterants, safflower and calendula, using chemometric tools. UV/Vis fingerprinting distinguished picrocrocin and crocin at λ_max_ = 230–260 nm and 400–470 nm, respectively, in saffron samples from other substitutes [25].

### 2.2. Caffeinated Beverages’ and Juices’ Quality Control Analyses 

The variation between plant species is either interspecific, resulting from a genetic basis, or intraspecific, related to agricultural practices and/or different cultivars. [26]. In this regard, the two harvests of *Camellia sinesis* tea leaf extracts were classified based upon their metabolite spectrum of UV-Vis light analysis, orthogonal signal correction, and partial least-squares discrimination analysis (OSC-PLS-DA) [27]. The calibration model was composed of 12 spectra for the first harvest and 14 from a second harvest, which was derived using OSC-PLS-DA in the discriminant spectral regions bands at 272, 410, and 663 nm among harvests. The 272 nm band corresponded to then n-π* of caffeine, particularly due to the chromophore C=O, whereas the bands at 410 and 663 nm were attributed to the presence of pheophytin, characterized by a blue region and Q bands. Both caffeine and pheophytin could serve as valued biomarkers for the QC of *C. sinesis* harvests and be used to distinguish different harvests in tea [27], but in other caffeine-rich drugs such as coffee, cocoa waits to be tested. 

Chang et al., 2016 developed a simple method for apple juice QC that can predict varieties’ type and adulteration with other fruit juices, sugars, and water, in addition to the quality of the juice and its age. In this study, the UV spectroscopic dataset was visualized using partial least-squares regression (PLS). The prediction of quality control attributes such as soluble solids, vitamin C, total phenolics, and reducing sugars was one goal of this work using PLSR-UV, with acceptable root mean square error (RMSE) ranging from 0.2555–2.3448, *R*^2^ = 0.7276–0.9816. Principal component regression (PCR-UV) was found adequate to predict pH and antioxidant activity with RMSE = 0.0000–2.7426 and *R*^2^ = 0.7073–1.0000. Furthermore, this study provided an insight into apple juice optimum storage time in a dark temperature-controlled storage locker (25 °C) for 7 days with RMSE = 0.4681 days and *R*^2^ = 0.9832 [28]. 

### 2.3. Dairy Products Quality Control Analysis

UV quantification was established by Xie et al. (2015) as a rapid and valid colorimetric assay to detect melamine (MEL) in milk samples using portable UV-Vis spectroscopy. The principle was based on a gold nanoparticle (AuNP) probe color change from red to purple after (MEL) induced aggregation of dispersed AuNPs, with results revealing that only within 15 min assay time were enough to record the amount of MEL in milk, with a minimal limit of 2 ppm [29]. 

Another application in dairy products analysis includes determining the binding effect of two different proteins called whey protein isolate and sodium caseinate with a hydrophobic carotenoid called lutein using UV/Vis spectroscopic analysis and circular dichroism (CD) spectroscopy. The turbidity of lutein and lutein–protein complex solutions was recorded by measuring transmission at 500 nm using UV/Vis. In contrast, CD spectra were recorded between 190 and 250 nm to predict the lutein binding effect on both protein structures that prove lutein’s chemical stability with improvement in its aqueous solubility. In this study, milk proteins protected lutein from decomposition and acted against oxidative power in a concentration-dependent manner. These results suggest that milk proteins could behave as protective carriers of lipophilic nutraceuticals and aid in their stabilization [30].

### 2.4. Honey Quality Control Analysis

Honey is known for its vast nutritional value acting as a superfood. Its rich and unique secondary metabolites’ bioactive composition includes polyphenols in addition to its rich sugar content [31]. Honey is continuously subjected to adulteration attempts especially for its rather expensive monofloral honey types such as ziziphus honey. Comparative UV fingerprinting of honey collected from Saudi Arabia of different origins showed efficiency in separating ziziphus honey from other types and suggests that UV spectroscopy coupled to chemometrics could aid in the authentication of genuine Saudi honey samples in a rapid, simple, and cheap way [32]. Whether the same platform can be used to authenticate other unifloral honey types such as Manuka honey from other origins has yet to be examined.

A comparative study targeting the authentication of 13 genuine monofloral Sidr honeys from different botanical origins was carried out using a UV fingerprint analyzed using PCA and soft independent modeling of class analogy (SIMCA). Generated models proved clear segregation of the genuine Sidr samples from other samples of low quality such as polyfloral or non-Sidr samples as, i.e., polyfloral with *Acacia tortilis*, *Acacia tortilis*, *Dracaena*, *Acacia mellefera*, and *Euphorbia* sp. Moreover, the SIMCA model revealed a clear demarcation of samples that could predict genuine Sidr honey samples mixed with lower price polyfloral honey within the limit of >10% [33].

### 2.5. Vegetable Oils: Quality Control, Authentication, and Adulteration Detection

Compared to several reports of UV targeting polar food matrices such as honey, etc., UV fingerprinting was also reported for the analysis of vegetable oils. One study was performed on olive, sunflower, soybean, and corn oils using UV spectroscopy (λ_max_ 300–540 nm) coupled to multivariate curve resolution with alternating least squares (MCR-ALS). This study was based on the fact that frying oils leads to the generation of oxidative compounds in addition to the degradation of the anti-oxidative ones. An antioxidant metabolite presented in oil “tocopherol” was evaluated after frying in addition to oxidation products generated with frying. MCR-ALS showed that tocopherol levels decreased concurrently with an increase in oxidative products at 110, 85, 70, and 50 °C for sunflower, colza, olive, soybean, and corn oils, respectively. Consequently, sunflower and olive oils were found more resistant to higher temperatures, compared to corn and soybean oils [34]. This study followed a previous study that recorded the highest level of *α*-tocopherol in sunflower oil followed by colza, olive, corn, and soybean oils [35]. 

A study to distinguish between virgin and extra virgin olive oil authentication was carried out on several virgin and extra virgin olive oils produced in Italy by determining their carotenoid and chlorophyll levels using two different UV/Vis absorption-based assays. The first assay defined two indexes: K670 revealed quantitative estimation of total chlorophylls at 670 nm while K470 provided total content of carotenoids at 470 nm. In contrast, the second assay depended on the mathematical deconvolution of the oils’ absorption spectra to determine levels of the significant pigments to include *β*-carotene, lutein, and pheophytin A/B. Satisfactory linear correlations between chlorophyll levels detected using the two assays were recorded in a relatively large concentration range, with linear regression (*R*^2^ = 0.9361) and a significant difference between virgin and extra virgin olive oils. Moreover, both methods showed a significant difference, with the recommendation of a second method to quantify total carotenoids and total chlorophylls (in non-fresh olive oil) versus the first method [36].

Didham et al. (2020) compared the capacity of two spectroscopic techniques, UV-Vis compared to attenuated total reflection (ATR-MIR), to detect adulteration in olive oil with other cheap vegetable oils, i.e., canola and sunflower. Results revealed that both techniques could detect adulteration in samples at a relative percentile level higher than 10% and fail to detect that of low concentration <5%. Differentiation between pure olive oil and vegetable oil adulterated samples mainly were attributed to differences in fatty acids, oxylipins, and pigments composition [37], suggesting that these are good markers for adulteration detection in olive oil if targeted using other chromatographic techniques, i.e., GC and HPLC. 

Green chemistry represents a relatively simple novel analytical approach that provides rapid information with no need for sample preparation, while additionally using less or no solvent. In this context, Garrido-Delgado et al. (2018) proposed a study using spectral fingerprint ultraviolet photoionization ion mobility spectrometry (UV-IMS) coupled with multivariate data analysis to detect adulteration of extra virgin olive oil by vegetable oils, i.e., corn, sunflower, and seed oils. In this study, 8 extra virgin olive oil samples were adulterated with different amounts of vegetable oils ranging from 10 to 50%, resulting in 96 adulterated samples submitted to other chemometric models. UV-IMS showed the ability to detect adulteration at a concentration ≥10% of lower price vegetable oils in time <20 min, including data processing. Additionally, partial least-squares regression (PLS), as a linear regression calibration method, recorded the quantity of vegetable oils adulterant added to the samples with a good regression (*R*^2^ > 0.72). Conclusively, UV-IMS could be used as a low-cost chemical fingerprint for olive oil authentication [38].

Likewise, Jiang, et al. (2015) performed a study using UV spectroscopic analysis coupled to PCA and PLSR to detect extra virgin olive oil (EVOO) samples adulterated with vegetable oils, i.e., soybean oil, corn oil, and sunflower oil. Results revealed that EVOO/vegetable oil adulterant, oil blends without EVOO, and an oil mixture with palm oil could be discriminated using PCA analysis. The results also showed that PLSR analysis depended on UV fingerprints for detecting EVOO and palm oil in oil blends, with low RMSE (0.001–0.710%) and higher *R*^2^ (0.853–1.000), suggestive for a valid model with no overfit. Consequently, UV spectroscopic analysis coupled to chemometric tools could discriminate EVOO–vegetable oil blends, oil blends without EVOO, and oil blends with palm oil [39].

### 2.6. Food Biowastes Valorization 

Lycopene is a major carotenoid in tomatoes with potential antioxidant, cardiovascular protective, and anticancer effects, especially against prostate cancer [40]. Recovery of lycopene from tomato waste products was assessed using UV fingerprinting to determine the influence of temperature and solvents’ compositional difference on extraction kinetics. Extraction was carried out using an acetone/hexane mixture at different ratios (1:3, 2:2, and 3:1, *v*/*v*) and temperatures of 30, 40, and 50 °C. Results revealed that temperature at 30 °C and a solvent ratio of 1:3 (*v*/*v*) were the most optimum for lycopene extraction, with experimental data and predicted values of yield similar to Peleg’s model, showing good accuracy, with an average root mean square error (RMSE) = 0.06213 and average mean bias error (MBE) = 0.00543 [40]. 

One UV application of food by-products concerned molasses as a valuable industrial by-product of sugarcane with several health benefits. Asikin et al. (2016) performed a study to identify antioxidant metabolites in sugarcane molasses and evaluate their DNA protection power against oxidative damage. Two metabolites were identified using bioassay and UV spectral monitoring, identified as 7,8-dehydrodiconiferyl alcohol-4-*O**-β*-d-glucoside and 7,8-simulanol-9′-*O**-β*-d-glucoside, with the latter compound found more active against free-radical-induced DNA damage. These results pose sugarcane molasses as a nutraceutical that can protect against oxidative damage [41].

Another application of UV for food valorization involved the estimation of yield and purity of piperine isolated from (*Piper nigrum* L.) industrial residue using UV, as well as spent pepper, after industrial processes. They were screened for active compounds, oleoresin, and piperine production from the residue, with optimization for extraction. Piperine content in oleoresin and crystallized piperine was estimated by UV at λ_max_ = 343 nm. UV spectra showed that piperine purities from raw (92.5%) and spent (93.6%) peppers were comparable to the result of standard piperine spectra. Such an approach confirmed the enrichment of oleoresin, piperine, and volatiles in both spent and raw pepper, revealing that spent pepper still encompasses 60% of the major bioactive compounds in pepper even after industrial processes [42]. A list of UV-based fingerprint applications in different food products’ and food biowastes’ valorization is presented in Figure 2 and further in Table 1, highlighting each application’s outcomes, advantages, and/or limitations.

## 3. UV Applications in Nutraceuticals/Functional Foods

The administration of herbal dietary supplements in markets has been on the rise, considering it is safe and has several health benefits compared to synthetic drugs. One problem concerning nutraceuticals QC lies in that active metabolites in plant extracts showed variation in context to climates, temperature, seasons, soils, and harvesting and processing methods, in addition to a potential to be contaminated with pesticides, heavy metals, mycotoxin, etc. [43,44].

The registration process of herbal pharmaceuticals requires the assessment of stability assays to confirm the quality of active metabolites under variable conditions and to aid in determining product shelf-life, as in the case of natural products [45]. Therefore, the authentication and quality control of phytoconstituents in herbal medicine and food supplements are crucial, and can be achieved using accurate analytical methods. Figure 3 summarizes UV-based fingerprint spectroscopic applications in nutraceuticals to include quality control, qualitative/quantitative assay, authentication/fingerprinting, and other drug applications. 

### 3.1. Quality Control and Authentication

Herbal products embrace both phytochemical complexity and interspecies variability, which are likely to readily affect the final product compared to synthetic drugs, with the latter being composed of one chemical, warranting better tools to overcome quality control challenges through setting up validated monographs (fingerprinting) and/or standardization processes [46].

Direct UV spectrometry compared to HPLC-UV was used to investigate ginger samples QC from different geographical origins with the aid of PCA. UV spectroscopy allowed for constructing spectroscopic profiles of authentic and marketed ginger samples showing two λ_maxes_ at 230 and 280 nm that showed maximum absorbance. The PCA model constructed using absorbance levels of these two λ maxes in specimens yielded no significant classification of authentic ginger samples according to their geographical origins, contrary to full-scan UV yielding better results concerning ginger quality control and authentication. A more advantageous application of HPLC-UV over direct UV spectrometry was the ability to determine storage shelf-life effects on pungent principle levels in ginger, i.e., 6-gingerol, 8-gingerol, 6-shagaol, and 8-shagaol. Ginger samples showed a significant decrease in quality level after storage for 3 years, establishing a validated PCA model to discriminate between samples before and after storage [47]. 

In search of substitutes for the rare *Swertia mileensis* in Chinese medicine, the other three *Swertia* species were compared using UV spectroscopy. *Swertia* species crude extracts exhibited two UV bands at 240 and 270 nm due to conjugated double bonds in secoiridoid. Further, UV absorbance of all *Swertia* species except *S. mileensis* shifted from 270 to 274 nm owing to a solid resonating structure consisting of C=O and benzene ring resulting in redshift in xanthonoids. PCA, alongside UV fingerprinting, revealed that the other three *Swertia* species were incompatible with *S. mileensis* metabolites’ profile, except for the presence of Swertimarin in all *Swertia* species detected at 250 nm [48]. 

Direct UV spectrometry was also employed for the assay of a common polyphenol used as a food supplement in preventive medicine, “naringin”. A major drawback for using this flavanone as a nutraceutical lies in its poor solubility and bioavailability, with only its aglycon “naringenin” found able to form a nanosized complex with a whey protein “*β*-Lactoglobulin”, with *K_a_* in the order of 10^4^ M^−1^. Results for the homogeneity of particle size, in addition to bonded nano complexes, observed by UV absorbance and a fluorescence quenching method, revealed that such a combination led to an enhancement in both solubility and bioavailability [49].

Glutathione is a 1^ry^ intracellular antioxidant that is involved in different biological actions, especially the role of the brain’s capacities to scavenge ROS with its depletion often associated with Parkinsonism pathogenesis [50]. Two validated analytical methods were established for detecting oxidative degradation of glutathione, a common antioxidant in nutraceuticals. The first method was based on fluorimetric detection of its reaction product with *o*-phthalaldehyde at *λ*_ex_/*λ*_em_ = 340/425 nm, whereas the second method was based on UV detection at 210 nm. Results revealed that oxidative mechanism by 3% H_2_O_2_ resulted in 95% oxidized glutathione within 180 min following pseudo 1st order kinetics [51].

*Penicillium chrysogenum* is a commonly occurring mold in indoor environments and foods and has gained much attention for its use in the production of the antibiotic penicillin [52]. Application of UV fingerprinting alongside chemometric tools attempted to predict cell viability of *P. chrysogenum*, reported to be a crucial step for the overall process performance. PCA modeling of the UV dataset aided further to identify nucleic acids and protein impurities (viability decline factors) at 260 and 280 nm, respectively. Models applied to predict *P. chrysogenum* viability all showed high predictive power of the UV datasets, showing normalized root mean squared error of cross-validation (NRMSEcv) for PLS, OPLS, and PCR = 0.05, 0.07, and 0.07 respectively [53]. 

Another use of UV-Vis coupled to chemometric tools, i.e., PLS-DA and HCA, is in fungal analysis. This study revealed discrimination between chloroform extracts of *Wolfiporia extensa* fungal species from different geographical origins, which showed low polar metabolites in *W. extensa* to serve as chemotaxonomic markers detected within UV range 190–450 nm with some common peaks in all samples at 287, 312, 326, and 340 nm. Two λ_max_ at 287 and 326 nm could be used for a quantitative test, as they are common and showed variable importance for the projection (VIP) greater than 1.0, which contributes to samples’ separation from PLS-DA [54].

### 3.2. Quantification-Based Assay

The core criteria for quality control assessment of natural products is conducting the appropriate and rigorous method to detect and quantify active agents for efficacy issues as typical in the case of targeted metabolites analyses. The employment of UV spectroscopy alone or in conjunction with chromatographic techniques has been extensively reported in the literature to achieve such a goal in the quantitative assay of natural products in nutraceuticals [2]. 

One UV-based quality control method was reported in the traditional Chinese medicine *Ginkgo biloba* tablet using spectral fingerprint quantification at λ_max_ 190–400 nm. This method is based on flowing injection analyses and a diode array detector that provides information on chemical nature based on chromophore characteristics from several indexes, i.e., spectral fingerprint, information, fluctuation, and information fluctuation. Analysis depended on fingerprint separation ratio (*β*) and fingerprint frequency (*ρ*) that meet requirements of fingerprint points numbers and a purity level of monochromatic light. Furthermore, fingerprint total signal intensity *LR* = 100,122–126,720, fingerprint *AUC* = 99,991–122,749, arithmetic mean absorbance *A* = 475–601, and geometric mean absorbance *A*_0_ = 392–497; overall, values referred to a UV spectrum higher response signal. Ginkgo tablets were evaluated for qualification grades, revealing that all batches were qualified (Grade ≤ 3) except for one batch due to its higher content. This approach revealed that a direct, simple, practical, and reliable quantified UV spectral fingerprint identification method by quantifications and digital recording of all chemical content chromophores characteristics was successful in the case of ginko and has yet to be applied in other traditional Chinese medicines [55].

## 4. Conclusions

UV spectroscopy is a simple, fast analytical tool compared to other spectroscopic tools, yet has the potential to be exploited for several applications. UV-Vis typically is recognized for its capacity to visualize metabolites with conjugated systems (chromophores) in different spectral regions, analyzing a wide array of metabolites such as flavonoids, phenolic acids, and pigments, posing it to be used in screening plant extracts before isolation attempts of a targeted metabolite. Focusing on the advantages of this technique as a non-destructive one allows further investigations of the samples to proceed. Its easy-to-use devices, minimal processing, and little training make it accessible for many labs. 

UV spectroscopy is suggested to provide chemical information about the complex system of nutraceuticals in a fast, simple, inexpensive, and reliable way, especially with the aid of chemometrics analytical platforms. Therefore, many applications of UV-Vis were developed in nutraceuticals and drug industries as exemplified in this review, including chemotaxonomy of species, authentication, fingerprinting, quantitative determination, and extended to diagnosis purposes. 

The application of UV fingerprinting in chemotaxonomic relationship investigation can aid to identify potential substitutes for genuine drugs in the market, as in the case of *Swertia mileensis* in Chinese medicine. UV-Vis represents a crucial role in detecting different contaminants in many spices, and coffee was also presented in this review. It can be extended to cover major spices subjected to fraud attempts. Furthermore, UV-Vis showed many applications to ensure food quality and safety, for example, in the field of superfood with nutritional value authentication, suggesting that UV spectroscopy coupled to chemometric tools could discriminate genuine honey samples from others, and distinguish between virgin and extra virgin olive oil. 

Therefore, UV-Vis is considered an applicable fast (few seconds), simple, accurate analytical tool that covers many areas for applications. Further, coupling of UV-Vis fingerprint with chemometric tools revealed data classification among specimens and discrimination between marker compounds in an untargeted manner. Some limitations are still facing UV-Vis spectrophotometry, including its low sensitivity and selectivity, so development of this spectroscopic tool is crucially needed, along with development of its potential uses in monitoring food processing steps and to ensure its consistency. 

## Figures and Tables

**Figure 1 foods-11-02867-f001:**
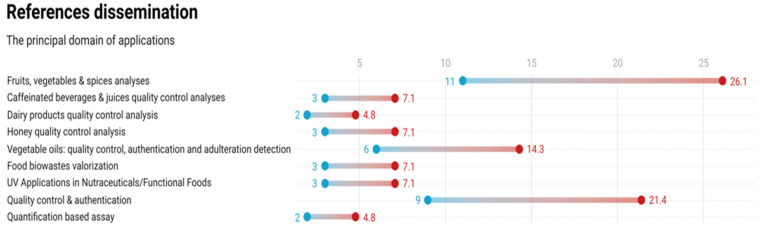
The principal domains of application of the UV-Vis fingerprinting approach and the corresponding references represented as number of references shown with blue circles and as percentages of total references with red circles.

**Figure 2 foods-11-02867-f002:**
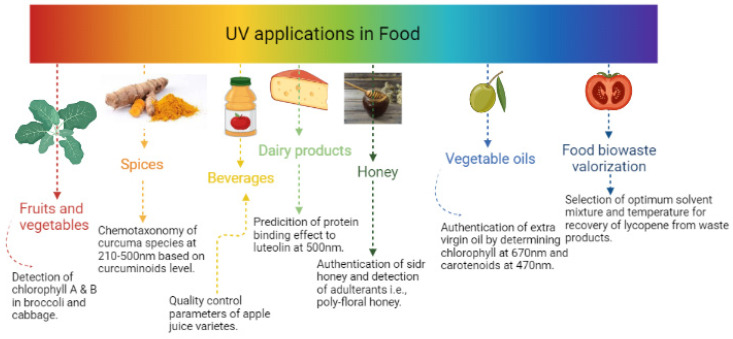
UV-Vis applications in different food products and food biowastes’ valorization.

**Figure 3 foods-11-02867-f003:**
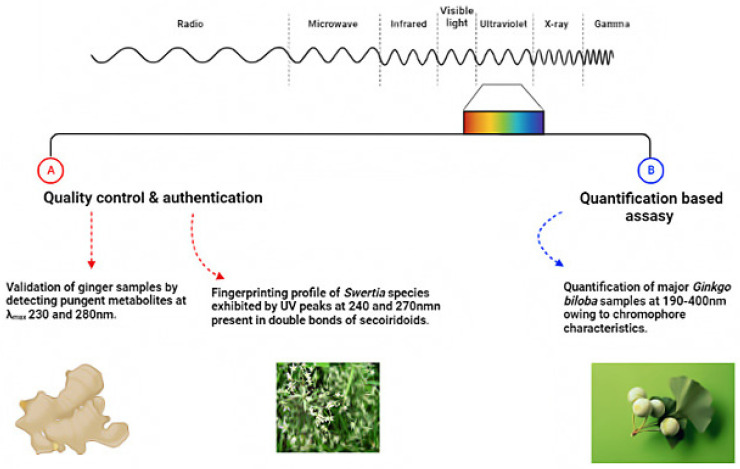
Quality control (**A**), qualitative/quantitative (**B**), of UV spectroscopy in the nutraceuticals industry.

**Table 1 foods-11-02867-t001:** List of major UV-Vis applications in the field of food production and analysis.

Application	Outcome	Advantages and/or Limitations	Ref.
Quantification of chlorophyll A and B in broccoli (*Brassica oleracea* Italica) and cabbage (*Brassica oleracea* Sabauda) as markers of the photosynthetic membrane using UV spectra coupled to partial least-squares regression (PLS).	-Fast alternative UV-chemometric tool for chlorophyll A and B determination in broccoli and cabbage plants.	-Limit of detection was at 0.174 and 0.304 μg mL^−1^, selectivity at 0.946 and 0.942, and sensitivity at 0.0324 and 0.0183 absorbance mL μg^−1^ for chlorophyll A and B, respectively. -Significant improvement in both accuracy and precision compared to HPLC.	[16]
Discrimination of different *Curcuma* Indonesian species: *C. longa, C. xanthorrhiza, C. aeruginosa*, and *C. mangga* based on UV-Vis spectra (210–500 nm) analyzed by PCA and DA.	-PCA showed segregation of *C. longa* from other species owing to high level of curcuminoids. -DA showed discrimination of *C. aeruginosa C. mangga* in the positive value of DF2.	-The developed model of DA allowed classification of species from the different samples with a 95.5% discrimination value.	[4]
Quality control of apple juice to predict varieties’ type, adulteration, and age using simple UV-Vis and PLS analytical method.	-Quality control parameters (soluble solids, vitamin C, total phenolics, and reducing sugars) were predicted using (PLSR-UV), with acceptable RMSE = 0.2555–2.3448 and *R*^2^ = 0.7276–0.9816. -Apple juice optimum storage time represented (RMSE) = 0.4681 day and *R*^2^ = 0.9832. -pH and antioxidant activity were predicted using PCR-UV with RMSE = 0.0000–2.7426 and *R*^2^ = 0.7073–1.0000.	-PCA-UV represented a potential portable tool for differentiating apple juice varieties, adulteration, quality, and ageing.	[28]
Detection of melamine (MEL) in milk samples using portable UV-Vis quantification colorimetric assay.	(MEL) Determination in milk samples based on gold nanoparticle (AuNP) probe color change from red to purple after MEL-induced aggregation of dispersed AuNPs.	-Short 15 min assay time -Minimal detection limit = 2 ppm.	[29]
Authentication of genuine honey samples using UV-Vis fingerprinting and chemometrics	-Authentication of 13 genuine monofloral Sidr honeys from low-quality polyfloral or non-Sidr samples.	-SIMCA model revealed clear demarcation of genuine Sidr honey samples from those mixed with lower price polyfloral honey within the limit of >10%	[33]
	-Efficiency in separating genuine Saudi Arabia ziziphus honey from other types of different origins.	-UV and chemometrics determined the authenticity of the botanical and geographical origin of Saudi honey in a rapid, simple, and cheap way	[32]
Olive oil authentication and adulteration detection	-Two different UV/Vis absorption assays to distinguish between virgin and extra virgin olive oils in Italy. The first assay defined two indexes: K670 and K470, which revealed quantitative estimation of total chlorophylls and carotenoids at 670 and 470 nm, respectively. The second assay depended on mathematical deconvolution oils’ absorption spectra determining levels of *β*-carotene, lutein, and pheophytin A/B.	-Both methods showed significant differences and linear regression (*R*^2^ = 0.9361), with the recommendation of the second method for the quantification of total carotenoids and total chlorophylls (in non-fresh olive oil) versus the first method.	[36]
	-Adulteration detection in olive oil with other cheap vegetables oil, i.e., canola and sunflower using UV-Vis assay representing oxylipids and pigments as good markers for olive oil adulteration detection.	-Adulteration detection in olive oil samples at relative percentile level higher than 10%, but failed to detect that of low concentration <5%.	[37]
	-UV photoionization ion mobility spectrometry (UV-IMS) and chemometrics revealed adulteration detection of eight extra virgin olive oils with corn, sunflower, and seed oils at a concentration ≥10% of lower price vegetable oils in time <20 min including data processing.	-Partial least-squares regression (PLS), as a linear regression calibration method, recorded the quantity of vegetable oils adulterant added to the samples with good regression (*R*^2^ > 0.72) revealing UV-IMS as a low-cost chemical fingerprint for olive oil authentication	[39]
Recovery of lycopene from tomato waste products using UV fingerprinting and parameters determination, i.e., temperature and solvent type for extraction optimization	Extraction using acetone/hexane mixture at different ratios (1:3, 2:2, and 3:1 *v/v*) and temperatures of 30, 40, and 50 °C revealed temperature at 30 °C and solvent ratio 1:3 (*v/v*) as the most optimum for lycopene.	Experimental data and predicted values of yield, showing good accuracy, with an average root mean square error (RMSE) = 0.06213 and average mean bias error (MBE) = 0.00543.	[40]
Estimation of yield and purity of piperine isolated from (*Piper nigrum* L.) industrial residue and spent pepper after industrial processes using UV and HPLC.	Piperine content estimated using abs. at λ_max_ = 343 nm showed piperine purity from raw (92.5%) and spent (93.6%) peppers were comparable to standard piperine spectra.	Rapid tool for determination of piperine level in pepper waste	[42]
Adulterants’ detection in market spices.	-UV-Vis coupled to PCA aided for the detection of adulterants such as rhodamine B and commercial red textile dye in chili commercial powder at 600 nm.	UV-Vis with PCA-DA represents fast, precise, and accurate authentication of chili powder from the two synthetics dyes.	[23].
	-UV-Vis coupled to PLS-DA detected synthetic dye adulterants, i.e., Sudan I or blends of Sudan I + IV dyes in culinary spices.	-Simple rapid method for culinary spices’ adulterant detection that generates hazardous metabolites and toxicity in the human body.	[24]

## Data Availability

Not applicable.

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
