# Peer review of "UV Fingerprinting Approaches for Quality Control Analyses of Food and Functional Food Coupled to Chemometrics: A Comprehensive Analysis of Novel Trends and Applications"

_foods, 2022, doi:10.3390/foods11182867_

Round 1

Reviewer 1 Report

The manuscript entitled “UV fingerprinting approaches for quality control analyses of food and functional food coupled to chemometrics: A comprehensive review of novel trends and applications” provides a review of the latest applications of the the UV-Vis spectroscopy in different area especially in the quality control of food and dietary supplements fields including also the most applied chemometric methods.

I appreciate the special work done for the collection and synthesis of the presented articles and also the good organization of this review - into representative chapters.

In my opinion some improvements are necessary.

General observations:

-                   What is not very clear and must be mentioned from the beginning (in the abstract) is the period to which this review refers - usually such articles include articles published in the last 10 years or more). Please clearly mention the period and specify if similar review was developed before in literature.

-                Please be very careful in the entire manuscript and correctly mention when refer to UV-specroscopy, UV-Vis spectroscopy and spectrophotometry (UV-Vis spectrometry) – some corrections were marked in the manuscript.

-                Please verify if all relevant papers were collected from literature in the reference period for every chapter. Some chapters (as UV Applications in Nutraceuticals/Functional Foods, Quantification based assay et all.) seems to present very few applications. The number of founded papers would be specified in each case (for every chapter).

-                If the number of applications is greater, the references can be presented in the form of a table (similar to table 1).

-                Some sentences (marked in the manuscript) are too long and difficult to understand. Please revise some of them.

-                For paper improvement, a short statistics (as for example the percent of publications for each chapter calculated based on total number of publications in the last years in the UV fingerprinting approaches – for example a Pie Chart graph that can be constructed in Statistica Software) would be realized. In conclusions chapter, specification of these percentages would be usefull.

-                More detail related to necessary corrections and suggestions are marked in the manuscript.

Reviewer 2 Report

please kindly revise followed my comment in attachment

Author Response

We have responded to all comments made within the pdf file sent by the reviewer and addressed comments in the submission word file

Reviewer 3 Report

In this paper, the authors describe the applications of UV-VIS spectra combined with chemometrics in food analysis. The topic is generally interesting; however major revisions are suggested before being considered for publication. Below are the comments:

- According to the title, the publication should concern quality control analysis of food and functional food coupled with chemometrics. However, some parts of the text do not relate to food analysis, sections 4, 5 and 6 (e.g. how forest soil monitoring is related to food analysis?) and do not match the publication as a whole. Therefore, in my opinion, you should remove them and additionally lines 140-142, 291-296 etc.

Minor concerns:

1. The abbreviation NP for "natural products" is misleading as it is often used in publications to denote nanoparticles. It would be better to avoid using this abbreviation concerning natural products.

2. "CLS: classical least squares" - is included in the abbreviations list twice.

3. LDA: Linear discriminate analysis – should be Linear discrimination analysis. Line 68 also needs correction.

4. RMSE: Root-mean-square deviation - should be Root mean square error. Line 147 also needs correction.

5. R2: linear regression - should be discrimination coefficient 

6. MCR-ALS add this to the abbreviations list

7. Unclear sentences which need correction:

Line 56: Multivariate regression methods, i.e., multivariate regression and multivariate calibration methods...

Lines 59-61: this sentence is unclear and clumsy

Line 84: ...analysis using UV spectra coupling(PLS)...

Lines 97-98: unnecessary italic 

Line 103: "...a maximum of 700 nm UV"?  - 700 nm is the visible range of the spectrum. Another issue is that carotenoids absorb in the 400-500 nm range.

 Line 116: "UV-Vis spectrophotometer coupled to PCA..."  PCA is not an instrument

Lines 157-160: these sentences require correction

In Table 1: using UV spectra coupled to  - unnecessary link to the website https://www.sciencedirect.com/topics/chemistry/ultraviolet-spectrum

In Table 1: DF2 - explain 

Line 278: two λ maxes

Line 300: unnecessary link to: https://www.sciencedirect.com/topics/food-science/beta-lactoglobulin

Round 2

Reviewer 1 Report

Figure 1 must to reveal the most important area of applications and the number and percentage (calculated as percent from the total number of papers found from these domains and cited as application references). For this, in the manuscript you can write as follow: “The principal domains of application, the number of publications for each domain and the corresponding percentage from the total number of publications in the UV-Vis fingerprinting area are presented in Figure 1.”

Also, Figure 1 must be corrected as follow:

1.      Write “The principal domains of application” instead of “References used among sections and subsections”

2.      Remove “ Introduction” -  the number and percentage of the papers  used in introduction is not relevant.

3.      In each case write the name of subsection and not “Subsection 2.1 ; Subsection 2.2   …” . For example instead of “Subsection 2.1” please write “ Fruits, vegetables & spices analyses” ; instead of  Subsection 2.2 please write “Caffeinated beverages & juices quality control analyses” et all.

4.      Revise the Figure Caption for figure 1. This would be as:” The principal domains of application of  the UV-Vis fingerprinting approach and the corresponding references represented as number of references showed with blue circles and as percentages of total references with red circles.”

If you removed the sections “UV Based Quality Control of Drugs” and “ UV Diagnostic Applications” , please be carefully to remove also the corresponding references (or include them in other sections).

Reviewer 3 Report

The manuscript has been significantly improved and I now recommend it for publication.

Author Response

we thank our reviewer for agreeing on the revised submission